



# Comparison of computational and experimental saturation vapor pressures of $\alpha$-pinene + O$_3$ oxidation products

Noora Hyttinen[1,2], Iida Pullinen[1], Aki Nissinen[1], Siegfried Schobesberger[1], Annele Virtanen[1], and Taina Yli-Juuti[1]

[1]Department of Applied Physics, University of Eastern Finland, P.O. Box 1627, FI-70211 Kuopio, Finland
[2]Now at: Department of Chemistry, Nanoscience Center, University of Jyväskylä, FI-40014 Jyväskylä, Finland

**Correspondence:** Noora Hyttinen (noora.x.hyttinen@jyu.fi)

**Abstract.** Accurate information on gas-to-particle partitioning is needed to model secondary organic aerosol formation. However, determining reliable saturation vapor pressures of atmospherically relevant multifunctional organic compounds is extremely difficult. We estimated saturation vapor pressures of $\alpha$-pinene ozonolysis derived secondary organic aerosol constituents using FIGAERO-CIMS experiments and COSMO-RS theory. We found a good agreement between experimental and

computational saturation vapor pressures for molecules with molar masses around 190 g mol$^{-1}$ and higher, most within a factor of 3 comparing the average of the experimental vapor pressures and the COSMO-RS estimate of the isomer closest to the experiments. Smaller molecules likely have saturation vapor pressures that are too high to be measured using our experimental setup. The molecules with molar masses below 190 g mol$^{-1}$ that have several orders of magnitude difference between the computational and experimental saturation vapor pressures observed in our experiments are likely products of thermal decom-

position occurring during thermal desorption. For example, dehydration and decarboxylation reactions are able to explain some of the discrepancies between measured and calculated saturation vapor pressures. Based on our estimates, FIGAERO-CIMS can best be used to determine saturation vapor pressures of compounds with low and extremely low volatilities.

## 1   Introduction

Secondary organic aerosol (SOA) is formed in the gas phase by the condensing of organic molecules with low volatilities. In

atmospheric science, organic compounds are often grouped based on their saturation vapor pressures to volatile organic compounds (VOC), intermediate volatility organic compounds (IVOC), semi-volatile organic compounds (SVOC), low volatility organic compounds (LVOC), extremely low volatility organic compounds (ELVOC) and ultra-low-volatility organic compounds (ULVOC) (Donahue et al., 2012; Schervish and Donahue, 2020). In the ambient air, ULVOCs can nucleate to initiate SOA formation (Kirkby et al., 2016; Bianchi et al., 2016), while ELVOCs, LVOCs and SVOCs can condense on existing parti-

cles to contribute to the growth of SOA (Ehn et al., 2014). A large source of organic compounds in the atmosphere are biogenic volatile organic compounds (BVOC) emitted by plants (Jimenez et al., 2009; Hallquist et al., 2009). These BVOCs are oxidized in the gas phase by oxidants, such as OH, O$_3$ and NO$_3$, to form less volatile compounds through addition of oxygen-containing





functional groups. In order to determine the role of different oxidation products on SOA formation, it is essential to have reliable methods to estimate the saturation vapor pressures of complex organic molecules formed in the atmosphere.

Monoterpenes ($C_{10}H_{16}$) are an abundant class of BVOCs emitted by various plants (Guenther et al., 1995). The oxidation mechanism of monoterpene reactants vary significantly, leading to products with different SOA formation capabilities (Thomsen et al., 2021). Additionally, the initial oxidant (i.e., OH, $O_3$ and $NO_3$) affects the SOA formation rates of the oxidation products (Kurtén et al., 2017). For example, $\alpha$-pinene, a very abundant monoterpene in the atmosphere, has been widely studied in both laboratory and field experiments (Docherty et al., 2005; Hall IV and Johnston, 2011; Hao et al., 2011; Ehn et al.,

2012, 2014; Kristensen et al., 2014; Lopez-Hilfiker et al., 2015; McVay et al., 2016; D'Ambro et al., 2018; Huang et al., 2018; Claflin et al., 2018; Ye et al., 2019), and $\alpha$-pinene oxidation, both with $O_3$ and OH, is efficient at producing oxygenated organic molecules and SOA. Consequently, the oxidation mechanism and potential structures of $\alpha$-pinene + $O_3$ products have been extensively studied both experimentally and computationally (Rissanen et al., 2015; Berndt et al., 2018; Kurtén et al., 2015; Iyer et al., 2021; Lignell et al., 2013; Aljawhary et al., 2016; Mutzel et al., 2016; Kristensen et al., 2020; Thomsen et al., 2021).

Various methods have been used for estimating the saturation vapor pressures or saturation mass concentrations of $\alpha$-pinene ozonolysis products (Ehn et al., 2014; Kurtén et al., 2016; D'Ambro et al., 2018; Buchholz et al., 2019; Peräkylä et al., 2020; Räty et al., 2021). However, measuring these saturation vapor pressures accurately is extremely difficult (Seinfeld and Pankow, 2003). Additionally, different experimental and theoretical methods are known to produce very different saturation vapor pressures (Bilde et al., 2015; Kurtén et al., 2016; Bannan et al., 2017; Wania et al., 2017; Ylisirniö et al., 2021). For example, the

agreement between different experiments is better measuring subcooled state compared to solid state, perhaps due to ambiguity of the physical state of the solid state samples (Bilde et al., 2015).

During recent years, saturation vapor pressures of atmospherically relevant multifunctional organics have been derived from their desorption temperatures using mass spectrometers equipped with the Filter Inlet for Gases and AEROsols (FIGAERO; Lopez-Hilfiker et al. (2014)). In this method, the saturation vapor pressures can be estimated based the desorption temperatures

of the molecules. However, the measurements need to be calibrated using compounds with known saturation vapor pressures in order to find the correlation between saturation vapor pressure and desorption temperature. For example, a recent experimental study highlighted how different sample preparation methods affect measured desorption temperatures of FIGAERO calibration experiments (Ylisirniö et al., 2021), and aerosol particle size and operational parameters generally affect the measurement results as well (Schobesberger et al., 2018; Thornton et al., 2020). Additionally, when the desorbed molecules are detected

using a chemical ionization mass spectrometer (CIMS), only the elemental compositions are obtained without any information on the chemical structures, based on which saturation vapor pressures could be further constrained.

Another possible complication in thermal desorption experiments is thermal decomposition reactions during the heating of the sample. For example, Stark et al. (2017) studied the effect of thermal decomposition on the determination of volatility distributions from FIGAERO-CIMS measurements. They concluded that most of the condense-phase species decompose during

thermal desorption experiments, in agreement with several other studies of laboratory and ambient FIGAERO measurements (Lopez-Hilfiker et al., 2015, 2016; Schobesberger et al., 2018). Most recently, Yang et al. (2021) found that decarboxylation





and dehydration reactions are significant in FIGAERO measurements for multifunctional carboxylic acids that have more than 4 oxygen atoms, degree of unsaturation between 2 and 4, and maximum desorption temperature ($T_{max}$) higher than 345 K.

Among theoretical models, the conductor-like screening model for real solvents (COSMO-RS; Klamt (1995); Klamt et al. (1998); Eckert and Klamt (2002)) has been seen as the most promising method for calculating partitioning properties, because it does not require calibration, unlike group-contribution methods (Wania et al., 2014). During the recent years, this quantum chemistry based method has been used to estimate saturation vapor pressures of atmospherically relevant multifunctional compounds (Wania et al., 2014, 2015; Kurtén et al., 2016; Wang et al., 2017; Krieger et al., 2018; Kurtén et al., 2018; D'Ambro et al., 2019; Hyttinen et al., 2020, 2021b). For example, Kurtén et al. (2016) compared COSMO-RS-derived saturation vapor pressures of 16 $\alpha$-pinene ozonolysis products with those estimated with various group-contribution methods. They found that COSMO-RS (parametrization BP_TZVPD_FINE_C30_1501 implemented in the COSMO*therm* program; COSMO*therm* (2015)) predicts up to 8 orders of magnitude higher saturation vapor pressures than group-contribution methods, such as EVAPORATION (Compernolle et al., 2011) and SIMPOL.1 (Pankow and Asher, 2008). The COSMO*therm*15-estimated saturation vapor pressures indicated that the studied highly oxidized monomers derived from the ozonolysis of $\alpha$-pinene were likely classified as SVOC with saturation vapor pressures higher than $10^{-5}$ Pa (Kurtén et al., 2016). However, the parametrization in COSMO*therm* has a large effect on the calculated properties, since the model is parametrized using a set of well-known compounds with experimental properties available. There have been significant improvements since the BP_TZVPD_FINE_C30_1501 parametrization used by Kurtén et al. (2016), especially with better description of the effect of hydrogen bonding on thermodynamic properties. This is an important factor in calculating properties of multifunctional compounds that are able to form intramolecular H-bonds. For example, Hyttinen et al. (2021b) found that with an improved conformer sampling method (recommended by Kurtén et al. (2018)) and a newer parametrization (BP_TZVPD_FINE_19), COSMO*therm*-estimated saturation vapor pressures of the two most highly oxygenated $\alpha$-pinene ozonolysis monomer products studied by Kurtén et al. (2016) are up to 2 orders of magnitude lower than SIMPOL.1 estimates, while COSMO*therm* predicted higher saturation vapor pressures than SIMPOL.1 for 15 different $\alpha$-pinene+OH-derived dimers.

In this study, we investigate the saturation vapor pressures of SOA constituents formed in $\alpha$-pinene ozonolysis, using both FIGAERO-CIMS experiments and the COSMO-RS theory. We compare saturation vapor pressures derived from both experiments and calculations (different isomers), in order to evaluate the experimental method. Additionally, we investigate the prevalence of thermal decomposition in our experiment.

## 2 Methods

### 2.1 Chamber Experiments

The experiments were conducted at a 9 m³ Teflon environmental reaction chamber. The chamber is located at University of Eastern Finland (Kuopio, Finland). During the experiment, the chamber was operated as a batch reactor, i.e., the experimental conditions were set at the start of the experiment, and after the chemistry was initiated, the proceeding changes in gas and particle phase in the close system were sampled. The chamber is set on a foldable frame which allows the chamber to collapse





when deflated, maintaining a constant pressure. The chamber and the instruments were situated inside a temperature-controlled environment (temperature set to 295.15 K). Before the experiment, the chamber was flushed over night with dry clean air, to reduce the impact of evaporation of residues from preceding experiments from the walls.

To prepare the chamber for the experiment, it was first filled with clean air, which was sampled by a proton-transfer-reaction time-of-flight mass spectrometer (PTR-ToF-MS, Ionicon, Inc.), and a Filter Inlet for Gases and AEROsols (FIGAERO) coupled
with a Time-of-Flight Chemical Ionization Mass Spectrometer (ToF CIMS) to determine chamber background. The next section will provide a more thorough description of the instruments. After the chamber was filled close to operational capacity (9 m$^3$), $\alpha$-pinene was introduced into the chamber. This was done by flushing dry purified air through an $\alpha$-pinene diffusion source and into the chamber until target concentration (11 ppb) was reached. $\alpha$-Pinene levels were monitored with an online PTR-ToF-MS. Polydisperse ammonium sulfate seed aerosol ($\sim$ 10 000 # cm$^{-3}$, maximum number concentration at $\sim$ 80 nm) was added to
provide condensation nuclei and to prevent possible nucleation during the experiments. Lastly, 30 ppb of externally generated ozone (using an ozone generator with UV lamp of wavelength 185 nm) was introduced into the reaction chamber to start the chemistry. Experiment duration was 8 hours from when the chemistry started (ozone was added). There was practically no change in the chamber size during the experiment, due to the low sampling flows compared to the total chamber volume.

### 2.1.1 Instrumentation

In this study, we analyzed particle-phase composition measurements performed with a Filter Inlet for Gases and AEROsols (FIGAERO) inlet system coupled with a time of flight chemical ionization mass spectrometer with iodide ionization (I-CIMS, Aerodyne Research Inc.), a system that allows for measurement of both gas-phase and particle-phase compounds with a single instrument (Lopez-Hilfiker et al., 2014, 2015; Ylisirniö et al., 2021). In the FIGAERO inlet, the aerosol particles are collected on a Teflon filter (Zefluor 2 $\mu$m PTFE Membrane filter, Pall Corp.) while simultaneously analysing gas phase. After a
predetermined collection time (here 45 minutes) is finished, the sampled particle matter is evaporated using a gradually heated nitrogen flow with a heating rate of 11.7 K min$^{-1}$ and the evaporated molecules are carried into the detector instrument I-CIMS. Integrating over the heating time will give the total signal of a particular compound in the sample being processed. The working principle of I-CIMS has been introduced elsewhere (Lee et al., 2014; Iyer et al., 2017), but in short, oxidized gas-phase constituents are detected by clustering negatively charged iodide anions (I$^-$) with suitable organic compounds. Clustering of
the organic molecules and I$^-$ happens in an Ion Molecule Reaction Chamber (IMR), which is actively controlled to be at 10$^4$ Pa pressure.

The particle sampling period was set to 45 minutes, and the particle analysis period consisted of 15 minutes ramping time (when the filter was heated linearly from room temperature to 473.15 K), and 15 minutes of soak period (where the filter temperature was kept at 473.15 K). Thus, there are 45 minutes of gas-phase measurements followed by a 30-minute gap while
particle chemical composition is being analysed. Seven particle samples were collected during the 8-hour SOA experiment.





### 2.1.2 Data analysis

All FIGAERO-CIMS data were preprocessed with tofTools (version 611) running in MATLAB R2019b (MATLAB, 2019), and further processed with custom MATLAB scripts. Saturation vapor pressures of the oxidized organics were estimated based on their thermograms, i.e., signal as a function of temperature along the heating of the particle sample in FIGAERO. We used 20 s averaging in the thermograms. The temperature axis calibration sample was made as described by Ylisirniö et al. (2021), by using an atomizer to produce a particle population with a similar size distribution to the one present in the chamber experiment. Polyethylene glycol (PEG$n$ with $n = 6, 7$ and 8) with known saturation vapor pressures (see Fig. S1 and Table S1 of the Supplement) were used to produce the calibration particle population. Following the calibration fit, the saturation vapor pressure ($p_{\mathrm{sat}}$ in Pa) of a molecule can be calculated from the temperature of the highest signal ($T_{\max}$ in K):

$$p_{\mathrm{sat}} = e^{-0.1594 \times T_{\max} + 40.13} \tag{1}$$

The variation in $T_{\max}$ values between 3 calibration runs varies from 0.5 K of the smallest PEG6 (282.3 g mol$^{-1}$) to 7.6 K of the largest PEG8 (370.4 g mol$^{-1}$). With our calibration curve, these differences correspond to a factor of 1.1 and 3.3 variation in the saturation vapor pressures, respectively. Saturation vapor pressures were calculated for multiple $\alpha$-pinene-derived SOA constituents from 6 different samples, i.e., 6 different subsequent thermal desorptions, during the one 8-hour experiment. The first sample of our experiment was omitted, because the signals were much lower in the first sample than the other samples. This was likely caused by lower concentrations of oxidation products in the chamber at the beginning of the experiment. In our experiment, the variation in $T_{\max}$ values between the different thermal desorption cycles ranged from 2.0 to 11.1 K. The variation in $T_{\max}$ values increases with the increasing molar mass (see Fig. S2 of the Supplement). The 11.1 K variation corresponds to a factor of 5.8 variation in $p_{\mathrm{sat}}$. Most of the studied compounds have saturation vapor pressures within a factor of 4 from the 6 measurement cycles.

### 2.2 COSMO*therm* calculations

Our experiments provided us with elemental compositions of compounds in our SOA sample and saturation vapor pressures corresponding to each composition. To compare with the experiments, we computed saturation vapor pressures of potential ozonolysis product structures corresponding to the measured elemental compositions using the COSMO-RS theory with the newest BP_TZVPD_FINE_21 parametrization, implemented in the COSMO*therm* program Release 2021 (BIOVIA COSMO*therm*, 2021). COSMO-RS uses statistical thermodynamics to predict properties of molecules in both condensed and gas phases. The interactions between molecules in the condensed phase are described using the partial charge surfaces of the molecules derived from quantum chemical calculations.

For our COSMO-RS calculations, we selected conformers containing no intramolecular H-bonds, detailed previously by Kurtén et al. (2018), Hyttinen and Prisle (2020) and Hyttinen et al. (2021b). This method has been shown to provide more reliable saturation vapor pressure estimates for multifunctional oxygenated organic compounds even if they are able to form intramolecular H-bonds (Kurtén et al., 2018). The conformer search was performed using the systematic algorithm (sparse





systematic algorithm for isomers that have more than 100 000 possible conformers) of Spartan'14 (Wavefunction Inc., 2014). Instead of omitting conformers containing intramolecular H-bonds after running the quantum chemical calculations, as recom-

mended by Kurtén et al. (2018), we removed conformers containing intramolecular H-bonds already after the initial conformer search step, in order to decrease the number of density functional theory (DFT) calculations needed for the input file generation (see Sect. S1 of the Supplement for the details).

The quantum chemical single-point calculations and geometry optimizations were performed using the COSMO*conf* program (BIOVIA COSMO*conf*, 2021), which utilizes the TURBOMOLE program package (TURBOMOLE, 2010). First, single-

point calculations at a low level of theory (BP/SV(P)-COSMO) were used to remove conformers with similar chemical potentials. After a geometry optimization at the same level, duplicate conformers with similar geometries and chemical potentials were omitted. Duplicates were also removed after the final optimization at the higher level of theory (BP/def-TZVP-COSMO) and final single-point energies were calculated for the remaining conformers at the highest level of theory available in the current COSMO*therm* version (BP/def2-TZVPD-FINE-COSMO, currently only available for single-point calculations). Fi-

nally, the intramolecular H-bonding of each remaining conformer was checked using the pr_steric keyword in COSMO*therm* and up to 40 conformers containing no intramolecular H-bonds were selected for our saturation vapor pressure calculations (see Kurtén et al. (2018) for more details). The gas-phase energies of the selected conformers were obtained by optimizing the condensed-phase geometries and calculating single-point energies at the levels of theory corresponding to the COSMO calculations (BP/def-TZVP-GAS and BP/def2-TZVPD-GAS, respectively). Gas-phase conformers containing intramolecu-

lar H-bonds (formed in the gas-phase geometry optimization) were omitted and the gas-phase single-point energy of the corresponding condensed-phase geometry was used instead. When all found conformers contained intramolecular H-bonds, conformers containing a single intramolecular H-bonds were selected for the COSMO-RS calculation (see Sect. S1 of the Supplement).

In COSMO*therm*, the saturation vapor pressure ($p_{\mathrm{sat},i}$) of a compound is estimated using the free energy difference of the

compound in the pure condensed phase ($G_i^{(l)}$) and in the gas phase ($G_i^{(g)}$):

$$p_{\mathrm{sat},i} = e^{-(G_i^{(l)} - G_i^{(g)})/RT} \tag{2}$$

Here $R$ is the gas constant and $T$ is the temperature.

## 3 Results and Discussion

### 3.1 Saturation vapor pressures

We selected 26 elemental compositions (20 monomers and 6 dimers) from our FIGAERO-CIMS measurements for the comparison with COSMO*therm*-estimated saturation vapor pressures. All elemental compositions that contain up to 10 carbon atoms are assumed to be monomers (containing carbon atoms only from the original reactant $\alpha$-pinene), while compounds with 11-20 carbon atoms are assumed to be dimers (covalently bound accretion products of two monomers). For COSMO*therm* analysis, we selected 1-7 isomer structures that can be formed from $\alpha$-pinene ozonolysis for each elemental composition. The degree of



unsaturation of the studied monomers and dimers is 1–4 and 4–5, respectively, determining how many double bonds or ring structures each isomer must contain.

The structures of the studied monomers were formed based on structures suggested by previous experimental and computational studies (Lignell et al., 2013; Aljawhary et al., 2016; Mutzel et al., 2016; Kristensen et al., 2014; Kurtén et al., 2015; Iyer et al., 2021). These structures are shown in Figs S3–S7 of the Supplement. For dimer calculations, we selected elemental

compositions that can be formed using the studied monomer structures, assuming a loss of either $H_2O_2$, $H_2O$ or no atoms from the original monomers. With a loss of $H_2O_2$ or $H_2O$, a dimer can be formed by recombination of two hydroperoxy or hydroxy groups to form a peroxide or an ether. Additionally, if one or both of the monomers are carboxylic acids, the dimer contains an ester or an acid anhydride (RC(=O)OC(=O)R') group, respectively. A dimer can also be formed in a condensed-phase reaction between a hydroxide and an aldehyde to form a hemiacetal (ROR'OH). In hemiacetal formation, no atoms are lost

from the reactant monomers. In order to reduce the number of computationally heavy dimer calculations, we selected only one pair of monomer isomers with the same elemental composition for each dimerization reaction. For most of the monomers used to form the studied dimers, the best agreement between experimental and computational saturation vapor pressures was found with the isomer that had the lowest COSMO*therm*-estimated $p_{sat}$. We therefore mainly chose the monomer isomers with the lowest $p_{sat}$ to form the studied dimer isomers. Table S3 of the Supplement shows, which monomers were used to form each of

the studied dimer isomers.

Figure 1 shows COSMO*therm*-estimated saturation vapor pressures of the studied isomers, as well as vapor pressures derived from the experimental $T_{max}$ values. The agreement between COSMO*therm*-estimated and experimentally determined saturation vapor pressures is good for molar masses higher than 190 g mol$^{-1}$. Even with a limited selection of dimer structures, the agreement between COSMO*therm* and FIGAERO-CIMS is very good, and even better agreement could likely be

found by selecting additional dimer isomers for COSMO*therm* calculations.

The large discrepancy between measured and calculated $p_{sat}$ of the lowest molar mass molecules (light gray bars in Fig. 1) suggests that the measured $T_{max}$ values are related to the thermal decomposition temperatures of larger compounds, rather than saturation vapor pressures of the measured elemental compositions. It is unlikely that COSMO*therm* would overestimate saturation vapor pressures by several orders of magnitude using the newest parametrization and improved conformer

selection (Kurtén et al., 2018). Additionally, if the low molar mass compounds are IVOCs ($p_{sat} > 10^{-2}$ Pa), as predicted by COSMO*therm*, they are not likely to contribute to the SOA formation. Conversely, the calibration curve sets a practical upper limit to experimentally derivable $p_{sat}$ based on the experiment temperature and premature evaporation. For example, the upper limit $p_{sat}$ corresponding to the initial temperature of the experiment ($T_{max} = 294.15$ K) is $1.2 \times 10^{-3}$ Pa. However, the highest experimental saturation vapor pressure among the studied molecules is $8.5 \times 10^{-6}$ Pa, which corresponds to $T_{max} = 325$

K. This may indicate that the SOA constituents selected for our analysis do not contain SVOCs and the selected elemental compositions corresponding to SVOCs in the experiments were in fact thermal decomposition products rather than oxidation products of $\alpha$-pinene ozonolysis.

Both the estimated and measured $p_{sat}$ values correlate with molar mass, the O:C ratio having only a small effect on the $p_{sat}$. Based on our FIGAERO-CIMS measurements, the studied monomer products derived from $\alpha$-pinene ozonolysis present





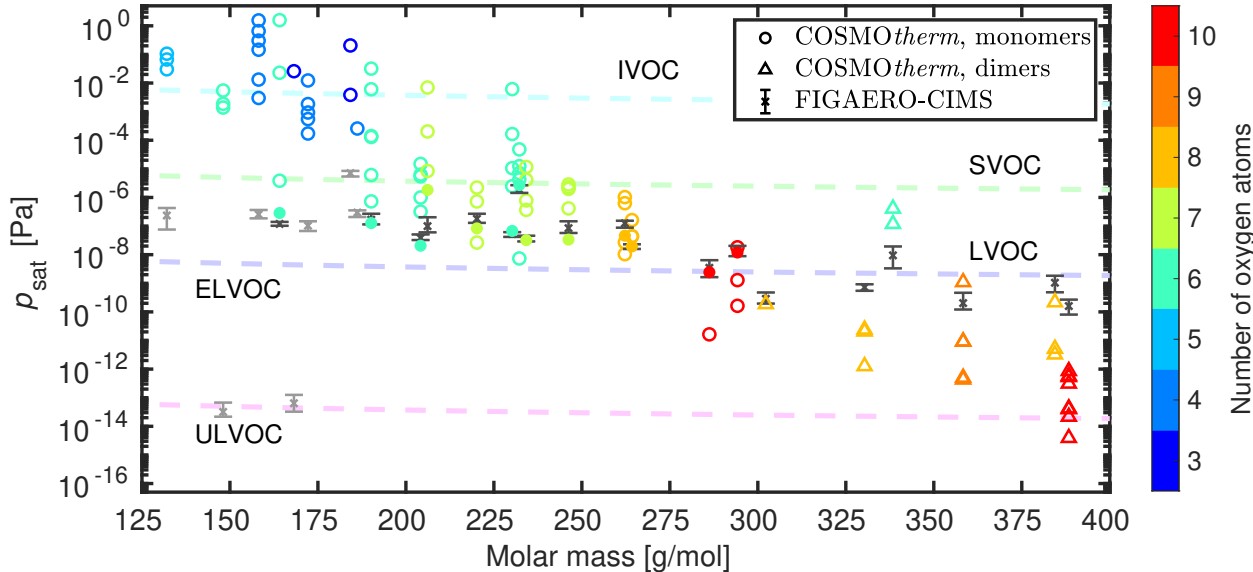

**Figure 1.** Saturation vapor pressures of the studied $\alpha$-pinene-derived ozonolysis products as a function of their molar mass at 298.15 K. The colors represent different number oxygen atoms in each isomer. The isomers shown with filled markers have COSMO*therm*-estimated $p_{\text{sat}}$ closest to the experiments. The bars of the experimental values show the range of saturation vapor pressures from the 6 used sample, instead of error estimates of the measurements. Suspected thermal decomposition products are shown with the lighter gray color. The dashed lines indicate different volatilities using the classification of Donahue et al. (2012) and Schervish and Donahue (2020) and assuming ideality ($\gamma = 1$) in the conversion from mass concentration to vapor pressure.

in the SOA are LVOCs or ELVOCs, while the studied dimers are mainly ELVOCs. We would like to note that this does not reflect the composition of $\alpha$-pinene ozonolysis SOA, but simply represents the set of elemental compositions selected for the analysis. The lowest experimental $p_{\text{sat}}$ among the studied elemental compositions is at $5.4 \times 10^{-11}$ Pa. The saturation vapor pressure corresponding to the upper limit temperature of our experiment ($T_{\text{max}} = 473.15$ K) is $4.7 \times 10^{-16}$ Pa, which means that saturation vapor pressures below $4.7 \times 10^{-16}$ Pa cannot be estimated in our experiments. It is also possible that saturation

vapor pressures of dimers with the lowest volatilities ($p_{\text{sat}} < 10^{-11}$ Pa) cannot be measured using thermal desorption, as the molecules would thermally decompose before evaporating from the sample (Yang et al., 2021).

      Recently, Hyttinen et al. (2021b) reported COSMO*therm*-estimated saturation vapor pressures of 15 different structural and stereo isomers of $\alpha$-pinene+OH-derived dimers with chemical composition $C_{20}H_{34}O_{10}$. They found saturation vapor pressures ranging from $3.4 \times 10^{-15}$ Pa (6 H-bond donors) to $1.0 \times 10^{-10}$ Pa (3 H-bond donors), with a strong correlation between $p_{\text{sat}}$

and the number of H-bond donors in the molecule. Comparison with our experimental saturation vapor pressure for $C_{20}H_{34}O_{10}$ ($3.7 \times 10^{-10}$–$1.3 \times 10^{-9}$ Pa) suggests that the isomers of $C_{20}H_{34}O_{10}$ detected by thermal desorption are likely to contain 3 H-bond donors rather than 6 H-bond donors. Isomers containing a higher number of H-bond donors (lower $p_{\text{sat}}$) are more likely to thermally decompose due to higher desorption temperatures and a higher number of functional groups that are required for thermal decomposition.





## 3.2 Correlation between monomer and dimer vapor pressures

The COSMO*therm* calculations of dimers are computationally more demanding than of monomers, due to larger size and higher number of possible conformers. In group-contribution methods, such as SIMPOL.1, saturation vapor pressure of a compound is estimated as the sum of contributions of each of the functional groups in the molecule:

$$\log_{10} p_{\text{sat},i} = \Sigma_k \nu_{k,i} b_k \tag{3}$$

We used the same approach to estimate the saturation vapor pressures of dimers and compared those values to saturation vapor pressures estimated using COSMO*therm*. However, instead of using the functional groups of the dimer, we used the contributions of the two monomers that formed the dimer. This way, the group-contribution term $b_k$ was replaced by COSMO*therm*-estimated saturation vapor pressures of the monomers multiplied with a scaling factor ($S_n$) to account for the changing functional groups and loss of atoms in dimerization reaction $n$.

$$p_{\text{sat,dimer}} = S_n p_{\text{sat,monomer1}} p_{\text{sat,monomer2}} \tag{4}$$

Many of the monomer isomers had to be altered slightly to accommodate the chosen dimerization reactions, which makes a direct comparison between monomers and dimers impossible. We therefore only investigate the acid anhydride formation, for which we have the most dimers to compare. For this comparison we formed additional $C_{13}H_{18}O_9$ dimers from all carboxylic acid isomers of $C_5H_8O_6$ and $C_8H_{12}O_4$, in order to better test the effect of molar mass on $S$.

Figure 2 shows the correlation between the COSMO*therm*-estimated saturation vapor pressures of dimers and of the monomers that were used to form the dimers. We see that the product of monomer vapor pressures is 1-3 orders of magnitude higher than the dimer vapor pressure. There is also a size dependence in the scaling factor, the values of $S$ as a function of dimer size are shown in Fig. S13 of the Supplement. Of the studied acid anhydride dimers, $C_{13}H_{18}O_9$ (the smallest dimer) isomers have scaling factors $10^{-3}$-$10^{-2}$ and $C_{17}H_{24}O_{10}$ (the largest dimer) isomers $10^{-2}$-$10^{-1}$. As a comparison, SIMPOL.1 predicts $S$ = $1.1 \times 10^{-2}$ for the acid anhydride (ketone and ester) formation from two carboxylic acid monomers, with no size dependence.

The correlation between COSMO*therm*-estimated saturation vapor pressures of monomers and dimers can be used to obtain rough saturation vapor pressures estimates of a larger number of dimer compounds by computing only the saturation vapor pressures of their constituent monomers. This reduces the computational cost of dimer calculations, since the number of conformers and the calculation times increase exponentially with the size of the molecule. For example, the COSMO*therm*-estimated saturation vapor pressures of the studied dimer with the highest molar mass ($C_{17}H_{24}O_{10}$) are much lower than the experimental ones, around a factor of 2 difference between the highest COSMO*therm* estimate and the lowest experimental value. Assuming that $C_{17}H_{24}O_{10}$ is an acid anhydride formed from $C_8H_{12}O_4$ and $C_9H_{14}O_7$, trying all combinations of the studied carboxylic acid isomers gives a $p_{\text{sat}}$ range of $5.6 \times 10^{-14}$–$7.0 \times 10^{-9}$ Pa using Eq. (4) and $S=(1.0\text{-}9.5) \times 10^{-2}$. This range overlaps with the experimental range of $8.1 \times 10^{-11}$–$2.7 \times 10^{-10}$ Pa (see Fig. 1).





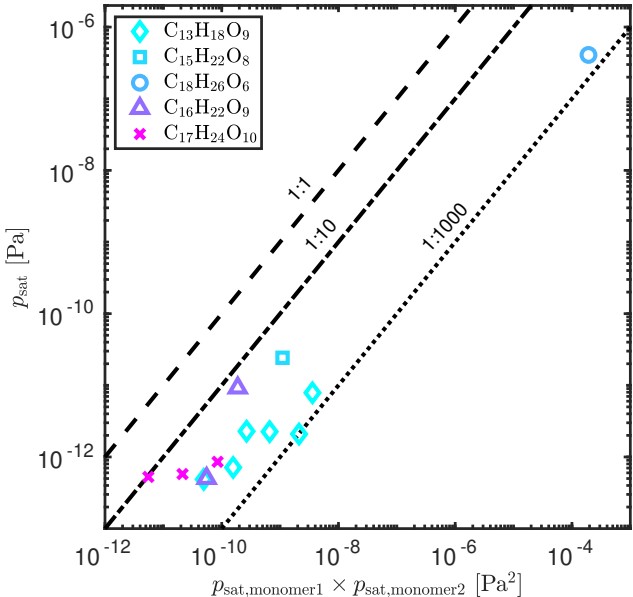

**Figure 2.** Correlation between COSMO*therm*-estimated saturation vapor pressures of the studied dimers ($p_{\text{sat}}$) and the product monomer vapor pressures ($p_{\text{sat,monomer}}$) at 298.15 K. The dimers are ordered from the smallest molar mass to the highest. The deviation from the 1:1 line represents the $S$ value of Eq. (4).

## 3.3 Thermal Decomposition

A recent study by Yang et al. (2021) proposed two major thermal decomposition pathways for multifunctional carboxylic acids occurring in FIGAERO-CIMS: dehydration (reaction 5) and decarboxylation (reaction 6).

$$\text{HOC(=O)RC(=O)OH} \rightarrow \text{R1C(=O)OC(=O)R1} + \text{H}_2\text{O} \tag{5}$$

$$\text{RC(=O)CH}_2\text{C(=O)OH} \rightarrow \text{RC(=O)CH}_3 + \text{CO}_2 \tag{6}$$

We selected 4 elemental compositions ($\text{C}_7\text{H}_{10}\text{O}_4$, $\text{C}_7\text{H}_{10}\text{O}_6$, $\text{C}_8\text{H}_{12}\text{O}_4$ and $\text{C}_8\text{H}_{12}\text{O}_6$) to investigate the two possible thermal decomposition reactions. The different isomers of $\text{C}_7\text{H}_{10}\text{O}_4$, $\text{C}_7\text{H}_{10}\text{O}_6$, $\text{C}_8\text{H}_{12}\text{O}_4$ and $\text{C}_8\text{H}_{12}\text{O}_6$, and their thermal decomposition reactants are shown in Figs S10 and S11 of the Supplement. These elemental compositions were selected, because their experimental and COSMO*therm*-estimated saturation vapor pressures had large differences. Additionally, their elemental compositions are possible products of both of the studied reactions. Both reactions are possible only if the product contains fewer than 10 carbon atoms (the monomer reactant of the dehydration reaction can contain up to 10 carbon atoms), and the degree of unsaturation is at least 3 (the product of dehydration contains at least one ring structure and two double bonds). The thermal





decomposition reactants also fulfill the number of oxygen and degree of unsaturation criteria given by Yang et al. (2021). Other likely decomposition products among the studied monomers are $C_4H_4O_5$, $C_4H_4O_6$, $C_9H_{12}O_3$, $C_{10}H_{16}O_3$ and $C_9H_{14}O_4$ (see Fig. 1).

Figure 3 shows the COSMO*therm*-estimated $p_{sat}$ of the studied thermal decomposition product isomers of $C_7H_{10}O_4$, $C_7H_{10}O_6$, $C_8H_{12}O_4$ and $C_8H_{12}O_6$ in red markers. The corresponding reactants are shown in blue markers at the product molar mass, and experimental $p_{sat}$ of the product elemental compositions are given as a range of the six measurement points. The studied thermal decomposition reaction is possible, if the COSMO*therm*-estimated saturation vapor pressure of the reactant molecule is lower than the measured saturation vapor pressure of the product elemental composition. Otherwise, the reactant
would desorb from the sample before the thermal decomposition reaction has taken place. For example, the elemental composition of $C_7H_{10}O_4$ at 158.15 g mol$^{-1}$ and its reactants $C_8H_{10}O_6$ (decarboxylation) and $C_7H_{12}O_5$ (dehydration) all have higher estimated saturation vapor pressures than the one derived experimentally (though Reactant-1 vapor pressure is close to the experimental one, see Fig. 3). This indicates that the measured $C_7H_{10}O_4$ is likely not a product of dehydration. The decarboxylation reaction is a possible source of the measured $C_7H_{10}O_4$, assuming under- or overestimation of the saturation vapor
pressure by our experiments or COSMO*therm*, respectively. Another possibility is that the measured $C_7H_{10}O_4$ is a fragmentation product of some other thermal decomposition reaction, where the reactant has an even lower saturation vapor pressure. For $C_8H_{12}O_4$, $C_7H_{10}O_6$ and $C_8H_{12}O_6$, some of the studied thermal decomposition reactants have saturation vapor pressures lower than the experimental $p_{sat}$. This means that the reactant molecules would remain in the sample at the measured $T_{max}$ (= potential thermal decomposition temperature). The measured $C_8H_{12}O_4$ is more likely a product of decarboxylation than dehydration,
because the proposed dehydration reactant has a higher COSMO*therm*-estimated saturation vapor pressure than the measured $p_{sat}$ of the product $C_8H_{12}O_4$. $C_7H_{10}O_6$ and $C_8H_{12}O_6$ have similar estimated and measured saturation vapor pressures and the measured molecules can therefore be either thermal decomposition products or simply relatively low-volatility isomers.

The saturation vapor pressures of the thermal decomposition reactant molecules are 3.3–6.5 (on average 4.7) orders of magnitude lower than the saturation vapor pressures of the corresponding product molecules. The difference between SIMPOL.1-
estimated saturation vapor pressures of the reactants and products are 4.9 and 3.9 orders of magnitude for the dehydration and decarboxylation reactions, respectively. Based on this, it is unlikely that the detected molecule is formed in either of these specific thermal decomposition reactions, if the COSMO*therm*-estimated $p_{sat}$ of the detected molecule is more than 7 orders of magnitude higher than the $p_{sat}$ derived from FIGAERO-CIMS experiments. In those cases, the reactant is likely a larger monomer or even a dimer, that decomposes to form two larger fragments.

It is also possible that other molecules detected in our FIGAERO-CIMS experiments are thermal decomposition products formed during the heating of the sample, though it is impossible to determine if this is true only based on information available from our measurements and calculations. If the decomposition temperature is lower than the $T_{max}$ of the decomposition product molecule, the measured $T_{max}$ values can correspond to the saturation vapor pressures of the decomposition products. However, this possibility was not taken into account when we selected the isomers for the COSMO*therm* calculations.





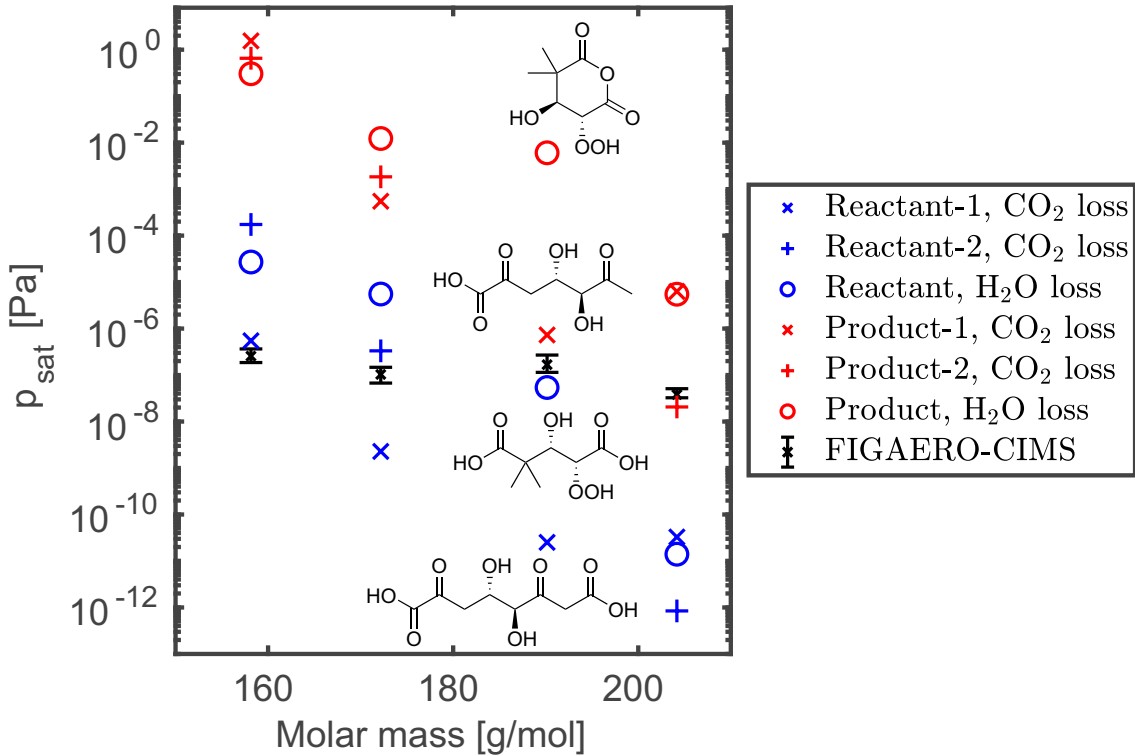

**Figure 3.** Saturation vapor pressures of potential products (red markers) and the reactants (blue markers) of thermal decomposition reactions (CO$_2$ loss: × or +, and H$_2$O loss: ○). The structures of the studied thermal decomposition reactants and products of C$_7$H$_{10}$O$_6$ are shown here as an example, the structures for the other 3 elemental compositions are shown in Figs S10 and S11 of the Supplement. Note that the reactant molecules are plotted with the same molar mass as the corresponding product molecule, instead of the molar mass of the reactant molecule.

### 3.4 Comparison with previous studies

Recently, Thomsen et al. (2021) identified multiple carboxylic acids in SOA formed in $\alpha$-pinene ozonolysis experiments using an ultra-high performance liquid chromatograph (UHPLC). Out of the compounds included in Thomsen et al. (2021), elemental compositions corresponding to diaterpenylic acid acetate (DTAA; C$_{10}$H$_{16}$O$_6$), 3-methyl-1,2,3-butanetricarboxylic acid (MBTCA; C$_8$H$_{12}$O$_6$), OH-pinonic acid (C$_{10}$H$_{16}$O$_4$), oxo-pinonic acid (C$_{10}$H$_{14}$O$_4$), pinonic acid (C$_{10}$H$_{14}$O$_3$) and terpenylic acid (C$_8$H$_{12}$O$_4$) were measured in our FIGAERO-CIMS experiments. In addition, an elemental composition corresponding to pinic acid (C$_9$H$_{14}$O$_4$) was seen in our experiments but we were not able to determine $T_{\max}$ values from the thermogram. Previously, Kurtén et al. (2016) calculated saturation vapor pressures of C$_{10}$H$_{16}$O$_4$, C$_{10}$H$_{16}$O$_6$ and C$_{10}$H$_{16}$O$_8$ with several isomers not included in our calculations using COSMO*therm*15. In Table 1, we have summarized saturation vapor pressures of the carboxylic acids identified by Thomsen et al. (2021), as well as all studied isomers of C$_{10}$H$_{16}$O$_4$, C$_{10}$H$_{16}$O$_6$ and C$_{10}$H$_{16}$O$_8$, from our COSMO*therm* calculations and FIGAERO-CIMS measurements, and previous studies. The experimental saturation





vapor pressures from previous studies (Bilde and Pandis, 2001; Lienhard et al., 2015; Babar et al., 2020) are given for the specific isomer, while COSMO*therm*15 values (Kurtén et al., 2016) are for various other isomers.

**Table 1.** Saturation vapor pressures of carboxylic acids in Pa. COSMO*therm*- and FIGAERO-CIMS-derived saturation vapor pressures are given at 298.15 K.

| Molecule name | Formula | $p_{\text{sat}}$, this study | | $p_{\text{sat}}$, previous studies | |
| --- | --- | --- | --- | --- | --- |
| | | FIGAERO-CIMS* | COSMO*therm*21 | experiments | COSMO*therm*15 |
| terebic acid | $C_7H_{10}O_4$ | $1.9\times10^{-7}$–$3.7\times10^{-7}$ | $3.0\times10^{-3}$ | - | - |
| terpenylic acid | $C_8H_{12}O_4$ | $6.7\times10^{-8}$–$1.5\times10^{-7}$ | $1.7\times10^{-4}$ | $(1.7\pm0.3)\times10^{-4\ c}$ | - |
| MBTCA | $C_8H_{12}O_6$ | $3.3\times10^{-8}$–$5.1\times10^{-8}$ | $3.2\times10^{-7}$ | $(1.4\pm0.5)\times10^{-6\ b}$ $(3.4\pm0.6)\times10^{-5\ c}$ $(2.2\pm1.6)\times10^{-8\ d}$ | - |
| pinic acid | $C_9H_{14}O_4$ | - | $2.6\times10^{-4}$ | $3.2\times10^{-5\ a}$ | - |
| pinonic acid | $C_{10}H_{16}O_3$ | $5.4\times10^{-6}$–$8.5\times10^{-6}$ | $3.9\times10^{-3}$ | $7.0\times10^{-5\ a}$ | - |
| OH-pinonic acid | $C_{10}H_{16}O_4$ | - | $1.0\times10^{-5}$ | - | $1.5\times10^{-2}$–$5.2\times10^{-2\ e}$ |
| DTAA other isomers | $C_{10}H_{16}O_6$ | $1.4\times10^{-6}$–$2.7\times10^{-6}$ | $1.7\times10^{-6}$ $7.4\times10^{-9}$–$4.8\times10^{-5}$ | $(1.8\pm0.2)\times10^{-5\ c}$ | $9.3\times10^{-4}$–$3.6\times10^{-2\ e}$ |
| various isomers | $C_{10}H_{16}O_8$ | $1.6\times10^{-8}$–$2.3\times10^{-8}$ | $1.9\times10^{-8}$–$1.6\times10^{-7}$ | - | $9.4\times10^{-5}$–$1.5\times10^{-2\ e}$ |

Experiments: [a] 296 K, Bilde and Pandis (2001), [b] 298.15 K, Lienhard et al. (2015), [c] 298.15 K, Babar et al. (2020), [d] 298 K, Kostenidou et al. (2018). COSMO*therm*: [e] 298.15 K, Kurtén et al. (2016) (different isomers). *The isomers detected in the FIGAERO-CIMS experiments may be different, or products of thermal decomposition.

Based on COSMO*therm*-estimated saturation vapor pressures, it is unlikely that the $T_{\max}$ values of pinic acid, pinonic acid, terebic acid and terpenylic acid could be determined in our FIGAERO-CIMS experiments due to their high volatilities. The FIGAERO-CIMS-derived saturation vapor pressures of $C_8H_{12}O_6$ and $C_{10}H_{16}O_6$ agree with previous measurements of MBTCA and DTAA by Babar et al. (2020) and Kostenidou et al. (2018), respectively. The 3 orders of magnitude difference in experimentally determined saturation vapor pressures of MBTCA (Lienhard et al., 2015; Babar et al., 2020; Kostenidou et al., 2018) demonstrates how different experimental methods give widely different values.

Kurtén et al. (2016) computed saturation vapor pressures of isomers that are potential products of gas-phase autoxidation, rather than products of condensed-phase reactions. The isomers in Kurtén et al. (2016) therefore contain mainly carbonyl, hydroperoxide and peroxy acid groups. Using a combination of group-contribution methods and COSMO*therm*15, they concluded that molecules with high oxygen content are likely LVOCs. However, systematic conformer sampling and newer COSMO*therm* parametrizations can lead to several orders of magnitude lower $p_{\text{sat}}$ estimates in COSMO*therm* (Hyttinen et al., 2021b). Two of the $C_{10}H_{16}O_6$ isomers studied here were taken from Kurtén et al. (2016). Our saturation vapor pressure estimates are 2-4 orders of magnitude lower than those estimated by Kurtén et al. (2016) (see Table S4 of the Supplement). Our calculations and experiments show that most of the studied dimers ($C_{13}$ and higher carbon numbers) are likely ELVOCs (around $p_{\text{sat}} < 10^{-9}$ Pa), the studied monomers with high molar masses (i.e., $C_9$-$C_{10}$ and $O_{10}$) may be ELVOCs, while the studied monomers with



lower molar masses (around $190 < M_w < 275$ g mol$^{-1}$) are likely LVOCs (around $10^{-9} < p_{\text{sat}} < 10^{-5}$ Pa), with the exception of some higher $p_{\text{sat}}$ isomers at lower molar masses ($M_w < 235$ g mol$^{-1}$).

## 4  Conclusions

We have shown that COSMO*therm*-estimated saturation vapor pressures agree (for $M_w > 190$ g mol$^{-1}$) with those derived from particle-phase thermal desorption measurements of the $\alpha$-pinene ozonolysis SOA system, taking into account the possibility of thermal decomposition. The measured $\alpha$-pinene ozonolysis monomer products selected from our SOA sample are mainly LVOCs and dimers are mainly ELVOCs. Molecules with ultra low volatilities are likely not desorbing during the experiments without fragmenting and are not detected by FIGAERO-CIMS. The smaller monomers ($M_w < 190$ g mol$^{-1}$) with the highest saturation vapor pressures (IVOCs) were likely not present in the sample aerosol collected from the chamber, instead, they are likely products of thermal decomposition formed from larger compounds during the experiment.

Comparison between estimated and experimental $p_{\text{sat}}$ can provide insight about the possible chemical structures of SOA constituents. Based on our results, the commonly used FIGAERO-CIMS instrument is best suited for measuring saturation vapor pressures of monoterpene-derived highly oxygenated monomers in the LVOC and ELVOC range with $M_w > 190$ g mol$^{-1}$. Hence, it is reliable for estimating saturation vapor pressures of oxidation products of monoterpenes, such as $\alpha$-pinene, keeping in mind that the smallest measured molecules are likely products of thermal decomposition. COSMO*therm* can be used to estimate saturation vapor pressures of compounds for which $p_{\text{sat}}$ is outside the applicable range of FIGAERO-CIMS experiments, i.e., IVOCs, SVOCs and ULVOCs, if the exact structures of the molecules are known.

In conclusion, this study gives us useful information for studying saturation vapor pressures of multifunctional compounds, and further on gas-to-particle partitioning of the compounds, which is the key when the SOA formation is investigated. Recently, it has been shown that SOA formation has a clear effect on both direct and indirect radiative forcing, highlighting the atmospheric relevance of our study.

*Data availability.* The research data have been deposited in a reliable public data repository (the CERN Zenodo service) and can be accessed at https://doi.org/10.5281/zenodo.5499485 (Hyttinen et al., 2021a).

*Author contributions.* SS, AV and TY designed the study, NH ran the COSMO-RS calculations, IP performed the experiments and wrote the experimental section, NH, IP and AN analyzed the data, NH, SS, AV and TY interpreted the results, NH wrote the manuscript with contribution from all coauthors.

*Competing interests.* The authors declare that they have no conflict of interest.



365 *Acknowledgement.* We thank Arttu Ylisirniö for his advice on the instrument calibration. We thank CSC - IT Center for Science, Finland, for computational resources and the tofTools team for providing tools for mass spectrometry analysis.

*Financial support.* This project has received funding from the Academy of Finland (grant nos 310682, 337550), and the European Union's Horizon 2020 research and innovation program, project FORCeS (grant no. 821205) and EUROCHAMP-2020 Infrastructure Activity (grant no. 730997).





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
