# Peer review of "Comparison of computational and experimental saturation vapor pressures of $\alpha$ -pinene + $O_3$ oxidation products"

_Atmospheric Chemistry and Physics, 2021_

## Author Response (AR1)

We thank both reviewers for their constructive comments, which have helped us improve the manuscript. Below is our detailed response to each point with the author reply in *italics* and the changes in the manuscript in **bold**. The line numbers are for the non-revised manuscript.

Referee #1

The manuscript by Hyttinen et al. examines the vapor pressures from α-pinene using both state-of-the-art measurement techniques and quantum mechanical simulations. The experiment and analysis are presented clearly and concisely. I recommend the article for publication and have a couple of clarifications to be addressed below.

Page 8 Figure 1: I can't entirely agree that the FIGAERO-CIMS is suited for the LVOC and ELVOC range. From my eye, it looks like the measurement and model data have two different slopes. Then their intercept where they happen to agree is between 225 and 300 g/mol. So I would be more inclined to agree if the slopes of the lines were in agreement for your stated "valid measurement range" and then in disagreement for the range outside.

*Author reply: We agree that the valid measurement range may be LVOC and the higher volatilities of the ELVOCs, since none of the higher molar mass molecules selected for our study have experimentally-determined saturation vapor pressures below $10^{-10}$ Pa. This has been clarified in the manuscript. Additionally, the difference between experiments and calculations in the dimers is explainable by the small number of dimer isomers that were included in the calculations (line 203: "Even with a limited selection of dimer structures, the agreement between COSMOtherm and FIGAERO-CIMS is very good, and even better agreement could likely be found by selecting additional dimer isomers for COSMOtherm calculations.").*

*If only the isomers that have COSMOtherm-estimated vapor pressures closest to the experiments were included in the figure, the slopes look quite similar (Figure R1) with a larger scatter in the COSMOtherm estimates in the dimers due to the relatively low number of studied dimer isomers.*

[Figure]

*Figure R1: Comparison of saturation vapor pressures between experiments and the isomer that has the COSMOtherm-estimated saturation vapor pressure closest to the experiment.*

**Changes in manuscript (line 12): Based on our estimates, FIGAERO-CIMS can best be used to determine saturation vapor pressures of compounds with low and extremely low volatilities at least down to $10^{-10}$ Pa in saturation vapor pressure.**

**(line 343): With our limited set of compounds, we cannot determine the lower limit saturation vapor pressure for which our experimental method is valid. Additionally, our limited set of calibration compounds further restricts our ability to reliably estimate saturation vapor pressures of the lowest volatility compounds. However, molecules with ultra-low volatilities are likely not evaporating from the sample during the experiments without fragmenting and are therefore not detected by FIGAERO-CIMS.**

Page 13 Table 1: Can you add the SIMPOL.1 values to the comparison table. Then add a sentence or two discussing those results and point to the SIMPOL.1 vs. COSMOtherm graph in the SI Figure S12 as you didn't mention that graph in the main text.

*Author reply: Thank you for this suggestion, we have added the SIMPOL.1 saturation vapor pressures to Table 1 and mention Figure S12 in the revised manuscript.*

**Changes in manuscript (line 339): We additionally compared our COSMO*therm* vapor pressures with those calculated with SIMPOL.1. The comparison is shown in Fig. S13 of the Supplement. We can see that with the molecules in this study, SIMPOL.1 is more likely to overestimate than underestimate COSMO*therm*-estimated saturation vapor pressures. COSMO*therm*-estimated saturation vapor pressures are up to factor 430 higher and up to factor 3.5x10$^4$ lower than those estimated using SIMPOL.1.**

**Table 1.** Saturation vapor pressures of carboxylic acids in Pa. COSMO*therm*-, FIGAERO-CIMS and SIMPOL.1-derived saturation vapor pressures are given at 298.15 K.

| Molecule name | Formula | $p_{\text{sat}}$, this study | | | $p_{\text{sat}}$, previous studies | |
|---|---|---|---|---|---|---|
| | | FIGAERO-CIMS* | COSMO*therm*21 | SIMPOL.1 | experiments | COSMO*therm*15 |
| terebic acid | $C_7H_{10}O_4$ | $1.9\times10^{-7}$–$3.7\times10^{-7}$ | $3.0\times10^{-3}$ | $1.5\times10^{-1}$ | - | - |
| terpenylic acid | $C_8H_{12}O_4$ | $6.7\times10^{-8}$–$1.5\times10^{-7}$ | $1.7\times10^{-4}$ | $5.5\times10^{-2}$ | $(1.7\pm0.3)\times10^{-4\ c}$ | - |
| MBTCA | $C_8H_{12}O_6$ | $3.3\times10^{-8}$–$5.1\times10^{-8}$ | $3.2\times10^{-7}$ | $8.4\times10^{-8}$ | $(1.4\pm0.5)\times10^{-6\ b}$ $(3.4\pm0.6)\times10^{-5\ c}$ $(2.2\pm1.6)\times10^{-8\ d}$ | - |
| pinic acid | $C_9H_{14}O_4$ | - | $2.6\times10^{-4}$ | $9.8\times10^{-5}$ | $3.2\times10^{-5\ a}$ | - |
| pinonic acid | $C_{10}H_{16}O_3$ | $5.4\times10^{-6}$–$8.5\times10^{-6}$ | $3.9\times10^{-3}$ | $1.4\times10^{-2}$ | $7.0\times10^{-5\ a}$ | - |
| OH-pinonic acid | $C_{10}H_{16}O_4$ | - | $1.0\times10^{-5}$ | $9.0\times10^{-5}$ | - | $1.5\times10^{-2}$–$5.2\times10^{-2\ e}$ |
| DTAA | $C_{10}H_{16}O_6$ | $1.4\times10^{-6}$–$2.7\times10^{-6}$ | $1.7\times10^{-6}$ | $2.5\times10^{-6}$ | $(1.8\pm0.2)\times10^{-5\ c}$ | $9.3\times10^{-4}$–$3.6\times10^{-2\ e}$ |
| other isomers | | | $7.4\times10^{-9}$–$4.8\times10^{-5}$ | $3.3\times10^{-7}$–$3.2\times10^{-3}$ | - | |
| various isomers | $C_{10}H_{16}O_8$ | $1.6\times10^{-8}$–$2.3\times10^{-8}$ | $1.9\times10^{-8}$–$1.6\times10^{-7}$ | $1.6\times10^{-9}$–$2.4\times10^{-4}$ | - | $9.4\times10^{-5}$–$1.5\times10^{-2\ e}$ |

Experiments: [a] 296 K, Bilde and Pandis (2001), [b] 298.15 K, Lienhard et al. (2015), [c] 298.15 K, Babar et al. (2020), [d] 298 K, Kostenidou et al. (2018). COSMO*therm*: [e] 298.15 K, Kurtén et al. (2016) (different isomers). *The isomers detected in the FIGAERO-CIMS experiments may be different, or products of thermal decomposition.

Referee #2

The data displayed in Figure 1 are interpreted by the authors as primarily indicating a large discrepancy between COSMOtherm-predicted vapour pressures and those obtained from the FIGAERO-CIMS technique for compounds with small molecular mass. However, an alternative reading of that plot would be that the two methods for obtaining vapor pressure indicate very different dependence of the vapour pressure on molecular mass. COSMOtherm suggests that with an increase in molecular mass by ~40 g/mol the vapor pressure decreases on average by ~1 order of magnitude. For the FIGAERO data that dependence is less than half as pronounced, whereby a three-fold increase in the molecular mass (from 130 to 390 g/mol) only lowers the vapor pressure from $10^{-7}$ Pa to $10^{-10}$ Pa. I note that the calibration involving PEGs (Table S1, Figure S1) indicates that an increase in the molecular mass of the PEGs by 88 g/mol leads to a 2.5 order of magnitude decrease in volatility, which is very comparable to what COSMOtherm suggests.

*Author reply: In addition to molar mass, saturation vapor pressure is known to depend on the functional groups of the molecule, as is seen in the several orders of magnitude difference in the COSMOtherm estimates of different isomers at the same elemental compositions. The fact that the PEG vapor pressures have such a clear correlation with molar mass is because each of them have the same 2 hydroxy groups and the size of the molecule increases in identical $C_2H_4O$ fragments. SOA constituents may well have a different slope between molar mass and $p_{sat}$ compared to PEG. We have previously noted that the addition of a $CH_2$ (~14 g/mol) to a multifunctional molecule has a 0.5 orders of magnitude effect on saturation vapor pressure and the addition of an oxygen atom (~16 g/mol) similarly has a 0.5-1 order of magnitude effect on saturation vapor pressure depending on the functional group (Hyttinen et al., 2021). This means that addition of an oxygen atom may decrease the saturation vapor pressure (Pa/(g/mol)) either less or more than the addition of a $CH_2$ group depending on the oxygen containing functional group. However, the COSMOtherm-estimated saturation vapor pressures can also vary more than an order of magnitude for different stereoisomers with identical functional groups (Kurtén et al., 2018).*

*Regarding the molar mass vs. $p_{sat}$ slope comparison between experimental and COSMOtherm derived values, please see our response to the first comment of Referee #1.*

In other words, the discrepancy between the two techniques is also quite large for the heaviest molecules – agreement is only apparent for molecules between 200 to 300 g/mol. The most likely explanation for the discrepancy is a limitation of the FIGAERO technique and not of the COSMOtherm

estimation method. The authors themselves offer an explanation on lines 224-226: "It is also possible that saturation vapor pressures of dimers with the lowest volatilities ($p_{sat} < 10^{-11}$ Pa) cannot be measured using thermal desorption, as the molecules would thermally decompose before evaporating from the sample (Yang et al., 2021)."

*Author reply: We agree that the agreement between experiments and calculations of the studied dimer isomers is not as good as of the monomers. However, the number of studied dimer isomers relative to all possible isomer is significantly lower due to the high number of possible dimers formed in both gas and condensed phase. It is therefore possible that the dimers detected in our experiments are isomers that have significantly different volatilities than the isomers that we used for our calculations.*

**Changes in manuscript (line 12): Based on our estimates, FIGAERO-CIMS can best be used to determine saturation vapor pressures of compounds with low and extremely low volatilities at least down to $10^{-10}$ Pa in saturation vapor pressure.**

At the root of the failure of the FIGAERO technique as applied here may be the extrapolation to volatilities that fall far outside of the calibration. The calibration only involves the molecular mass range from 282 to 370 g/mol and log $p_{Sat}$ from -5 to -7, but it is applied to a mass range from 130 to 390 g/mol and a vapour range of more than 10 orders of magnitude.

*Author reply: The calibration has been investigated in a previous publication by Ylisirniö et al. (2021) and also PEG5 was included in that fit. As the desorption temperature of PEG5 was already below 25C, calibration compounds with even higher saturation vapor pressures could not be included because of the temperature of the experiments. As the reviewer correctly noted, the extrapolation to lower saturation vapor pressure may be problematic, since reliable calibration curves have not been determined using compounds with extremely low, known saturation vapor pressures. However, in addition to a simply linear extrapolation towards lower log($p_{sat}$), Ylisirniö et al. (2021) proposed an alternative, polynomial extrapolation based on $T_{max}$ observations for higher-order PEG (n = 8 to 16), following the assumption that their log($p_{sat}$) decrease linearly. Accordingly, we added rough estimates for how much the experimental saturation vapor pressures might decrease if the calibration curve was polynomial instead of linear, as was proposed by Ylisirniö et al. (2021).*

**Changes to manuscript (line 131): Ylisirniö et al. (2021) found a good exponential correlation between temperature of the highest signal and saturation vapor pressure ranging up to $p_{sat}$=5x10$^{-4}$ Pa (PEG5). However, like theirs, our calibration only reached down to 9x10$^{-8}$ Pa in saturation vapor pressure, which introduces an additional source of uncertainty to the saturation vapor**

pressures estimated from the experiments. In addition to the linear correlation between $T_{\text{max}}$ and $\log_{10}p_{\text{sat}}$, Ylisirniö et al. (2021) proposed a polynomial calibration curve, which leads to lower saturation vapor pressure estimates at higher desorption temperatures ($T_{\text{max}} > 350$ K). With our 3 calibration points, it is impossible to find a reliable polynomial fit to extrapolate to higher $T_{\text{max}}$. Instead, we assume a similar difference between the two calibration curves to what was estimated by Ylisirniö et al. (2021). For example, using the linear fit of this study, 392 K corresponds to $2\times10^{-10}$ Pa, but in the polynomial fit, the same $T_{\text{max}}$ corresponds to about $10^{-11}$ Pa (see Table S2 and Figure S2 of the Supplement for more values).

Table S2: Difference in $T_{\text{max}}$ between linear (lin) and polynomial (pol) calibration fit in Ylisirniö et al.,[a] linear calibration fit of this study. The estimate for the polynomial fit for calibration data from this study was calculated by subtracting the $\Delta T_{\text{max}}$ from the linear calibration fit.

| $p_{\text{sat}}$ (Pa) | $T^a_{\text{max,pol}}$ (K) | $T^a_{\text{max,lin}}$ (K) | $\Delta T_{\text{max}}$ (K) | $T^b_{\text{max,lin}}$ (K) | $T^b_{\text{max,pol}}$ (K) |
|---|---|---|---|---|---|
| $3.6\times10^{-5}$ | 306 | 305 | -1 | 316 | 317 |
| $1.6\times10^{-6}$ | 324 | 323 | -1 | 335 | 337 |
| $9.0\times10^{-8}$ | 339 | 340 | 1 | 354 | 353 |
| $3.9\times10^{-9}$ | 352 | 358 | 6 | 373 | 368 |
| $2.0\times10^{-10}$ | 364 | 375 | 12 | 392 | 380 |
| $9.7\times10^{-12}$ | 373 | 393 | 20 | 411 | 391 |
| $4.3\times10^{-13}$ | 382 | 411 | 29 | 430 | 401 |
| $2.2\times10^{-14}$ | 389 | 428 | 39 | 449 | 410 |
| $1.0\times10^{-15}$ | 396 | 446 | 51 | 468 | 418 |
| $5.1\times10^{-17}$ | 401 | 464 | 63 | 487 | 424 |
| $2.4\times10^{-18}$ | 405 | 481 | 77 | 506 | 429 |

[a]Ylisirniö et al.,[S2] [b]This study.

[Figure]

Figure S2: Different calibration curves shown in Table S2.

**(line 205): Using a polynomial correlation between $T_{max}$ and $\log_{10}p_{sat}$, the $p_{sat}$ estimates of the studied monomers (highest $T_{max}$ at 378 K) would likely decrease by 1 order of magnitude or less (see Fig. S2 of the Supplement). With such a small decrease, all of the studied monomers would still be classified as LVOCs, with the exception of $C_9H_{18}O_{10}$, which would be classified as ELVOC. The experimental saturation vapor pressures of the studied dimers (excluding $C_{18}H_{26}O_6$) would decrease by 1 to 2 orders of magnitude, which would improve the agreement between the experimental and calculated saturation vapor pressures. Until more accurate calibration of the FIGAERO-CIMS instrument becomes available, the experimental $p_{sat}$ from the linear and polynomial fits can be used as upper and rough lower limit estimates, respectively.**

I suggest some rephrasing of formulations in abstract and manuscript text would be called for, e.g. line 4 to 5: "We found a good agreement between experimental and computational saturation vapor pressures for molecules with molar masses around 190 g mol$^{-1}$ and higher." Maybe the thrust of the paper could be shifted towards highlighting how the calibration procedure AND thermal decomposition limits the FIGAERO technique to a estimating the volatility to certain set of compounds and that there is both an upper and a lower volatility limit to this technique.

*Author reply: We think that the agreement is good between the experimental and computational saturation vapor pressures, even at the high molar masses given the small number of dimer isomers included in the calculations. An explanation on why the agreement with the dimers is not as good as the monomers is explained on lines 203-205. However, we have specified that the lowest saturation vapor pressures are likely not accessible with the FIGAERO-CIMS.*

**Changes in manuscript (line 82): We are especially interested in whether the calibration done using compounds with saturation vapor pressures limited to the LVOC/SVOC range is valid for estimating saturation vapor pressures of ELVOCs and ULVOCs.**

I even wonder whether it is really appropriate to refer to the values obtain by the FIGAERO technique as "experimental saturation vapour pressures" (as done in the title of the manuscript). This is a rather indirect inference based on a number of assumptions that may not be all that valid. Maybe formulating as "vapour pressure values inferred from regressions with desorption temperatures" would be more cautious.

*Author reply: We think that the word "experimental" does not necessarily mean that we are measuring the quantity in question. In our case, the saturation vapor pressures are derived from experimental methods, as opposed to purely theoretical calculations. The phrase "measured $p_{sat}$" was changed into "experimental $p_{sat}$" in the manuscript, since what was measured is $T_{max}$ and the $p_{sat}$ was estimated from the $T_{max}$ using a calibration curve. To be more specific and avoid confusion, we added the method names in the title.*

**Changes to manuscript (title): Comparison of saturation vapor pressures of α-pinene + O₃ oxidation products derived from COSMO-RS computations and thermal desorption experiments**

Line 24: Just because the SOA community has been using saturation vapour pressure does not mean that "it is essential to have reliable methods to estimate the saturation vapor pressures of complex organic molecules formed in the atmosphere". What really is required is the equilibrium partitioning ratio between the SOA and the gas phase. The saturation vapour pressure is simply commonly used to estimate that partitioning ratio (together with an activity coefficient of the compound in the SOA). A better formulation would be therefore: "it is essential to have reliable methods to estimate the volatility of complex organic molecules formed in the atmosphere."

*Author reply: Thank you for this suggestion, we have changed the manuscript accordingly.*

Line 43ff.: The thermodynamic property controlling the rate or timing of desorption would only be the saturation vapour pressure, if the molecule were to desorb from its own pure (liquid) phase. This doesn't appear to be the case in the method referred to here. As such, this method does not "measure" the saturation vapour pressure, but the equilibrium partitioning ratio. Would it then not be better to calibrate the method using compounds with known equilibrium partitioning ratios in order to find the correlation between equilibrium partitioning ratios and desorption temperature? After all, it is the equilibrium partitioning ratios that you are interested in in the first place.

*Author reply: Without knowledge on the exact condensed phase composition it is impossible to determine the activity coefficients of the SOA constituents. Here we assume that the compounds in the aerosol are similar enough to estimate the mixture as a pure ideal solution with activity coefficient equal to unity. In reality, the activity coefficients of the molecules studied here are likely between 1 and 10, based on activity coefficient calculations of multifunctional compounds in water insoluble organic matter (Hyttinen et al., 2021). This would lead to maximum one order of magnitude additional uncertainty to the experimental saturation vapor pressures.*

**Changes in manuscript (line 141): We used desorption temperatures to estimate saturation vapor pressures even though the particle-to-gas partitioning in our experiment is also affected by the activity coefficient of the compound in the sample. For example, Ylisirniö et al. (2021) found a 5-7 K difference in the temperatures of maximum desorption signal between pure PEG and PEG-400 mixture (average molecular mass ~ 400 g/mol), which they attributed to the additional compounds in the mixture. In the case of similar multifunctional compounds, the activity coefficients of individual compounds in the mixture (estimated using COSMO*therm*) are likely to be close to unity, with respect to pure compound reference state (the compound has similar chemical potentials in pure state and in the mixture, which leads to activity coefficient close to 1 in COSMO*therm* calculations). We therefore assume that the mixture is ideal and estimate saturation vapor pressures from desorption temperatures.**

Line 149ff.: I appreciate that you refer to the earlier study by Kurten et al., 2018 to justify the selection of conformers containing no intramolecular H-bonds, but I think some sort of explanation would still be required here. There doesn't seem to be any compelling reason why the ozonolysis of a-pinene should preferentially lead to oxidation products that do not contain intramolecular H-bonds. Furthermore, one of the advantages of a method such as COSMOtherm is precisely the fact that it should be able to account for the effect of intramolecular H-bonding on solvation. The rationale provided, namely "This method has been shown to provide more reliable saturation vapor pressure estimates for multifunctional oxygenated organic compounds", seems not very convincing as it could simply be coincidence.

*Author reply: The question is not as much which type of conformers are formed in the ozonolysis of α-pinene, since the energy barriers between different conformers are low and the molecule can switch between different conformers depending on its surroundings. In a vacuum, which is generally assumed in gas-phase calculations, the conformers that contain multiple intramolecular H-bonds are energetically most favorable, while in a polar solution, conformers that contain fewer intramolecular H-bonds are more stable, as they can interact with the surrounding solvent. We have added more explanation about the conformer selection in COSMOtherm calculations in the manuscript. In theory, the conformer distribution should be calculated accurately in COSMOtherm by including all conformers to the conformer set. However, we have noticed that when different types of conformers are included in the calculation, the conformer distribution in the condensed phase leads to worse agreement with experimentally derived condensed-phase properties and a better agreement is found when the appropriate conformers are selected for the calculation (Hyttinen and Prisle, 2020).*

**Changes to the manuscript (line 152): Additionally, Hyttinen and Prisle (2020) found that in COSMO*therm,* conformers containing multiple intramolecular H-bonds are given high weights in the conformer distribution due to their low COSMO energies, even if conformers containing no intramolecular H-bonds would be more stable in the condensed phase.**

Line 96ff.: Doesn't that procedure lead to a bias in the comparison of measured and estimated saturation vapour pressure?

*Author reply: We are not sure what the reviewer is referring to. The experimental set-up described in line 96 onwards is a standard way of conducting these types of experiments in a batch reactor chamber. We believe there might have been a typo in the line number but we're not able to determine to which procedure the reviewer is referring based on the comment.*

Line 15: "grouped […] **in**to"
Line 23: "the role […] **in** SOA formation"
Line 44: "estimated **from** the desorption temperatures"
Line 131: "**for** the smallest" "**for** the largest"

*Author reply: Thank you for these corrections, they were all changed in the manuscript.*

**References**

Hyttinen, N., Wolf, M., Rissanen, M. P., Ehn, M., Peräkylä, O., Kurtén, T., and Prisle, N. L.: Gas-to-Particle Partitioning of Cyclohexene- and α-Pinene-Derived Highly Oxygenated Dimers Evaluated Using COSMO*therm*, J. Phys. Chem. A, 125, 3726–3738, 2021.

Hyttinen, N. and Prisle, N. L.: Improving Solubility and Activity Estimates of Multifunctional Atmospheric Organics by Selecting Conformers in COSMO*therm*, J. Phys. Chem. A, 124, 4801–4812, https://doi.org/10.1021/acs.jpca.0c04285, 2020.

Kurtén, T., Hyttinen, N., D'Ambro, E. L., Thornton, J., and Prisle, N. L.: Estimating the saturation vapor pressures of isoprene oxidation products C5H12O6 and C5H10O6 using COSMO-RS, Atmos. Chem. Phys., 18, 17 589–17 600, https://doi.org/10.5194/acp-18-17589-2018, 2018.

Ylisirniö, A., Barreira, L. M. F., Pullinen, I., Buchholz, A., Jayne, J., Krechmer, J. E., Worsnop, D. R., Virtanen, A., and Schobesberger, S.: On the calibration of FIGAERO-ToF-CIMS: importance and impact of calibrant delivery for the particle-phase calibration, Atmos. Meas. Tech., 14, 355–367, https://doi.org/10.5194/amt-14-355-2021, 2021.

---

## Author Response (AR2)

We thank the editor for the additional comments. Below is our detailed response to each point with the author reply in *italics* and the changes in the manuscript in **bold**. The line numbers are for the revised manuscript with tracked changes.

Editor comments (line numbers refer to the track-change version of the manuscript)

Referee #2, comment 1 ("The data displayed in Figure 1 are interpreted by the authors as primarily indicating a large discrepancy..."): Your response to this comment was very detailed and convincing. Please add some of it to the manuscript.

*Author reply: Thank you for this suggestion. We have added some text to the section where we discuss the dependence of molar mass on saturation vapor pressure.*

**Changes to manuscript (line 250): In addition to molar mass, saturation vapor pressure is known to depend on the functional groups of the molecule, as is seen in the several orders of magnitude difference in the COSMO*therm* estimates of different isomers at the same elemental compositions. We have previously noted that the addition of a $CH_2$ (~ 14 g mol$^{-1}$) to a multifunctional molecule has a 0.5 orders of magnitude effect on saturation vapor pressure and the addition of an oxygen atom (~ 16 g mol$^{-1}$) similarly has a 0.5-1 order of magnitude effect on saturation vapor pressure depending on the functional group (Hyttinen et al., 2021b). This means that addition of an oxygen atom may decrease the saturation vapor pressure (in Pa per g mol$^{-1}$) either less or more than the addition of a $CH_2$ group depending on the oxygen containing functional group. The COSMO*therm*-estimated saturation vapor pressures can also vary more than an order of magnitude for different stereoisomers with identical functional groups (Kurtén et al., 2018).**

l. 28: mechanism --> mechanisms
*Author reply: This was changed in the manuscript.*

l. 42: ..is better when measuring at (or 'in a') subcooled state...
*Author reply: This was changed in the manuscript.*

l. 46: In this method --> Using this method
*Author reply: This was changed in the manuscript.*

l. 56: condense-phase --> condensed-phase

*Author reply: This was changed in the manuscript.*

l. 142: 'of this study' – does 'this' refer to here to the current study or to 'this previously mentioned study by Ylisirniö et al.?

*Author reply: By "this study", we mean the current study. This was clarified in the manuscript*

**Changes to manuscript: For example, using the our linear fit, 392 K corresponds to 2x10$^{-10}$ Pa, ...**

l. 201: Please add units to the parameters that are defined for the equation

**Changes to manuscript: In COSMO*therm*, the saturation vapor pressure ($p_{\text{sat},i}$ in mbar) of a compound is estimated using the free energy difference of the compound in the pure condensed phase ($G_i^{(l)}$ in kcal mol$^{-1}$) and in the gas phase ($G_i^{(g)}$ in kcal mol$^{-1}$):**

**Here $R$ is the gas constant (in kcal K$^{-1}$ mol$^{-1}$) and $T$ is the temperature (in K).**

l. 337: 'larger' than what, than the reactant? Or simply 'two large fragments'?

*Author reply: Yes, we meant "large fragments", this was changed in the manuscript.*

**References**

Hyttinen, N., Wolf, M., Rissanen, M. P., Ehn, M., Peräkylä, O., Kurtén, T., and Prisle, N. L.: Gas-to-Particle Partitioning of Cyclohexene- and α-Pinene-Derived Highly Oxygenated Dimers Evaluated Using COSMO*therm*, J. Phys. Chem. A, 125, 3726–3738, 2021b.

Kurtén, T., Hyttinen, N., D'Ambro, E. L., Thornton, J., and Prisle, N. L.: Estimating the saturation vapor pressures of isoprene oxidation products C5H12O6 and C5H10O6 using COSMO-RS, Atmos. Chem. Phys., 18, 17 589–17 600, https://doi.org/10.5194/acp-18-17589-2018, 2018.